# Rising food prices and poverty in Pakistan

**Nadia Shabnam** [1] *, **Neelam Aurangzeb**[2], **Salma Riaz**[3]

**1** Department of Health Professions Education, National University of Medical Sciences, Rawalpindi, Pakistan, **2** Department of Mathematics and Statistics, International Islamic University, Islamabad, Pakistan, **3** Department of Applied Mathematics and Statistics, Institute of Space Technology, Islamabad, Pakistan

* nadia.shabnam@numspak.edu.pk

## Abstract

An upsurge in global food prices in 2008 led to significantly higher food prices across the developing world. Global commodity prices have since declined but still remain volatile, but at the same time local food prices remain high in many countries. This study examines the potential impacts of the rise in food prices on poverty—income based poverty and calorie-based poverty- focusing on Pakistan, and its rural and urban areas. For this purpose, we used HIES data collected in three waves 2005–06, 2007–08 and 2010–11. Price elasticities are computed using binary Logistic regression method. The study results show that price of wheat, rice, milk, meat, fruit, pulses appear to distinguish the status of a household. Price elasticities shows that urban households are hit harder than rural households in calorie-poverty model. Overall, rising food prices are likely to lead higher poverty in Pakistan, as the negative impact on net consumers outweighs the benefits to producers. Therefore, effective strategy for eliminating poverty is far more concerned with price increases. Safety net programs can be more effective, but geographic targeting and other investments to strengthen safety nets are necessary to ensure that fewer people are affected by future crises. Government policies oriented towards relieving the food price pressure on the Pakistani poor should aim at lowering the prices of wheat, rice, eggs, oil, milk, and chicken.

**Data Availability Statement:** The data underlying the results presented in the study are available from https://www.pbs.gov.pk/content/microdata.

## Introduction

Poverty is the key root of hunger and malnutrition as well as denial of different economic and social aspects of life and fulfillment of human potential. Eradicating poverty in all its forms remains one of the greatest challenge facing humanity. While the number of people living in extreme poverty dropped by more than half between 1990 to 2015, too many are still struggling for the most basic human needs. The phenomenal hikes in food prices around the world had irrefutably devastating effect on the world's poor. According to World Bank report 2009, the price of major staple food such as wheat and rice rose by 121% and 76% in 2007–08. As a result of the food price crisis around 65 million people were estimated to survive under the $2 per day poverty line in 2009 [1]. Food and Agriculture organization (FAO) 2009 also reported that, the proportion of hungry people is approximately 15% (963 million) of the estimated world population [2]. In most developing countries, specifically in Sub-Saharan Africa and South Asian countries, poverty is spreading all over. These dramatic surges in food prices

**Funding:** The author(s) received no specific funding for this work.

**Competing interests:** The authors have declared that no competing interests exist.

increasing the difficulties of these countries and grind down the gains of poverty reduction made during last decade.

Rising world prices have different pass-through effects on domestic prices depending on the extent of protection, subsidies, the relative shares of domestic consumption met by imports, and domestic market structures. While these pass-through effects vary considerably, it is clear that rapidly increasing global commodity prices led to higher domestic food prices and contributed to social unrest across all continents which includes Uzbekistan, Mexico, Pakistan, and Cameroon [3]. The boom in domestic food prices could have long-term negative effects, even in countries where this was temporary. There are clear links between higher prices, lower caloric intake, lower quality diet, and an increase in child malnutrition. In light of the risks of serious long-term consequences of even temporary spikes in the price of essentials, the manner and extent to which the poor are insulated matters a great deal [3].

This paper focuses on Pakistan, a developing country with per capita GDP of only US$ 914.73 in 2008. After attaining high economic growth (6.8%) in 2006–07, Pakistan has seen its rate of growth decline considerably–to 3.7% in 2007–08 and 1.2% in 2008–09, respectively. Inflation increased in 2007–08 and touched its high peak (23.7%) in 2008–09 [4]. The net impact of rising food prices is different across households and income groups, since average Pakistani household spends about half of its income on food. Higher food prices almost always reduce the welfare of urban households because they are net purchasers of food. In contrast, most rural households produce some food items, so the effect of changing food prices on their welfare will depend on whether they are net purchasers or net sellers of food.

Regarding poverty status in Pakistan, no uni-directional movement has been observed in percentage of people living below poverty line. The head count ratio of 30.6% in 1998–99, increased to 34.5% in 2000–01, before declining to 23.9% and 22.3% during 2004–05 and 2005–06 [5]. Poverty is mostly measured in monetary terms, obtained by levels of income or consumption of per capita of a household. Based on income approach, the past century has seen many improvements in global prosperity and tried to get more people moving out of poverty state. However, Pakistan has also made significant progress in poverty reduction and improving human development over the past three decades but with slower rate.

In Pakistan, construction of poverty line is calorie based. Expenditure on calorie intake of 2350 kilo calories per day per adult equivalent, holding consumption expenditure on non-food items is aggregated to compute a poverty line. This poverty line is adjusted at the time of the poverty estimation after accounting for the inflationary impact in intervening years. According to a study by the UN interagency Assessment Mission (2008), the number of people in Pakistan with inadequate food consumption, less than 2,100 kilocalories per capita per day, rose from 72 million (45% of the total population) in 2005–06 to 84 million (51%) in 2008 as a result of increasing food prices. Thus, food price inflation has resulted in increasing food insecurity in the country [6,7]. On the other side, income poverty has declined in Pakistan over the period of 2005–06 to 2010–11 while calorie poverty has infect increased. Remarkably, the results are sensitive to calorie norms and poverty measure.

Several researchers have studied the impact of higher food prices on poverty in low-income countries. Deaton (1989) used nonparametric methods to examine the impact of a hypothetical change in rice prices on Thailand's income distribution and found that higher rice prices benefit all rural households, but especially middle-income households [8]. Ravallion and van der Walle (1991) report that a 10% increase in food prices raised the rate of poverty in Indonesia [9]. Also using nonparametric techniques, Barrett and Dorosh (1996), observed negative impacts of higher rice prices on the welfare of the rural poor in Madagascar because the gains to net rice sellers were concentrated among higher income rice farmers [10]. Ivanic and Martin examined nine low-income countries and concluded that increased staple food prices

would increase poverty in most, but not all, of those countries [11]. Vu and Glewwe explore the impact of rising food prices on poverty in Vietnam and observed that average welfare was found to increase because the average welfare loss of households whose welfare declined (net purchasers) was smaller than the average welfare gains of those whose welfare increased (net sellers) [12]. Similarly, many studies exist in literature from Pakistan [13] addressed the effects of rising food and fuel costs on poverty head count in Pakistan and found that a 20% increase in food prices would lead to an 8% increase in the poverty head count. Haq et al., considered the unexpected component of higher domestic food prices in 2007/2008, own and cross price compensated elasticities are used to derive the changes in the quantity consumed, food expenditure and impacts on poverty assuming the food crisis happened in 2004/2005. The results indicate that overall poverty increased by 34.8%, severely affecting the urban areas where poverty increased by 44.6% as compared to 32.5% in rural areas [14]. Shabnam et al., (2016) examined the impact of food price crises on the nutrient demand and concluded that food security deteriorated because of the food price crisis [7]. Azid et al., concluded that rising food prices is more an issue of supply rather than demand. Different studies show different mechanism through which the rise in prices affects the individuals' living standard. We expand it more and contribute to the literature to provide the extent the impact of domestic food price increase on poverty, including many food groups instead of only primary food commodities [15].

The Purpose of this paper is to cover the overlap between income poverty and calorie poverty as well as to examine the impact of rising food prices on expenditure and calorie poverty, trace out the determinant of expenditure and calorie poverty, and whether an increase in food price affect the poverty in urban and rural Pakistan over time. As mentioned above, the income poverty line is determined in accordance to a norm for calorie intakes. Even in absolute terms head count ratio of calorie poverty are far higher than income poverty. It may seem obvious that nutritional requirements vary not only with climate, but also with gender, age and activity status. Taste has changed significantly and food basket has become more diversified over time. Therefore, there is need to reassess the nutritional norms separately for urban and rural areas in particular, and for overall in general. Information on the effect of food price increase on poverty gives better policymaking on food security and poverty alleviation. Although, in this paper, we do not consider the positive impact of rising food prices on food-producing households.

The remainder of this article is organized in 5 sections: section 2 describes the data and methodology. Section 3 highlights the descriptive results to assess poverty incidence, and presents the elasticities and partial effects of variables from logit model. Section 4 includes the conclusions and section 5 contains policy implications and recommendations.

## Data description and methodology

For our analysis, we use the Pakistan Social and Living Standard Measurement Survey (PSLM) for the period of three years, 2005–06, 2007–08 and 2010–11 [16–18]. Household Integrated and Economic Survey (HIES) is the most important part of PSLM which is used for the computation of monetary poverty. The HIES includes the consumption module that provides the information on the quantities and values of 69 food items, however it does not provide the information on quantities of nonfood items but only expenditures. The food consumption module of the HIES provides the main data for the estimation of food/calorie poverty and expenditure poverty. The HIES data are cross-sectional in nature that is geographically stratified, cluster random sample of households across Pakistan. In addition to the information on household food, nonfood expenditure and demographic attributes, the HIES is having detailed

questions on other household characteristics such as occupation and education level of household head, health and immunization and other environmental characteristics. To compute the calorie consumption quantity from the reported food quantities, we used the conversion factors from Food Composition Table for Pakistan *(GOP Revised 2001)* [19]. The method of extraction of calories from different food groups is given in detail in [20].

## Poverty measures

The government of Pakistan (GOP) publishes the official poverty line using the HIES data based on estimated per adult equivalent monthly expenditure associated with consumption of a given minimum calorie requirement of 2350 per adult equivalent. Commonly the poverty status of a household is to be measured by the household aggregate consumption expenditure that is considered to be a welfare indicator. Once the poverty line is defined by the government, the household poverty status can be classified by relating this poverty cut point to the household consumption expenditure. For this purpose, we need to aggregate this information into a single index that could be used to describe the status of a group of households. There are several measures to compute this index, but these measures are sensitive to distribution among the poor [21].

We compute this index for income poverty and calorie poverty by using the well-known Foster Greer-thorbecke (FGT) class of indices [22]. According to FGT, these measures can be defined as

$$P(\alpha) = \frac{1}{H} \sum_{h=1}^{i} \left[ \frac{\lambda - \lambda_i}{\lambda} \right]^{\alpha} \tag{1}$$

$$P(\alpha) = \frac{1}{H} \sum_{h=1}^{i} \left[ \frac{\theta}{\lambda} \right]^{\alpha} \quad \text{where } \theta = \lambda - \lambda_i \tag{2}$$

Where $\theta$ is the gap between poverty line $\lambda$ and $\lambda_i$ income or expenditure of a poor household $i$ in case of income poverty and similarly it is also the gap between the calorie norm $\lambda$ and $\lambda_i$ calorie intake of poor household $i$, $i$ is the number of poor households defined as $\lambda_i < \lambda$; H is the total number of households in the sample.

In the above FGT index $P(\alpha)$ is measure of poverty and by putting the different values of $\alpha$ such as $\alpha = 0$, 1 and 2 provides different poverty measures that relates different weights to the degree of poverty and as well as inequality among the poor people. By putting the value of $\alpha = 0$, the poverty measure $P(0)$ collapse to the poverty incidence or headcount ratio of poor people, that is proportion of households below the poverty line thus, referred to as head count ratio (HCR). When $\alpha = 1$, then Poverty measure $P(1)$ is referred to as poverty gap index (PGI) or depth of poverty. It measures the average distance from poverty line. PGI is equal to the HCR multiplied by the average gap between the poverty line and income or calorie consumption of poor household, expressed as a percentage of the poverty line. When $\alpha = 2$, $P(2)$ measures the severity of poverty. This poverty gap squared takes into account the degree of inequality among poor households and gives higher weight that are far below the poverty line.

## The model and choice of explanatory variables

A household's status of falling in a poverty condition or not is modeled as a binary logit problem. The probability of a household being in poverty is

$$P(C_i = 1) = \frac{\exp(a_i + b_i \ln p_j + c_i \ln y_{ij} + d_i Z_{ij} + \mu_{ij})}{1 + \exp(a_i + b_i \ln p_j + c_i \ln y_{ij} + d_i Z_{ij} + \mu_{ij})} \tag{3}$$

$$P(I_i = 1) = \frac{\exp(a_i + b_i \ln p_j + c_i Z_{ij} + \mu_{ij})}{1 + \exp(a_i + b_i \ln p_j + c_i Z_{ij} + \mu_{ij})} \tag{4}$$

Where $i$ = ith household, $j$ = cluster of residence, $p_j$ = vector of food prices, $y_{ij}$ = household income and $Z$ = Vector of household characteristics, involving age, education, family size and its composition and employment etc. and $\mu$ = disturbance term.

In the above equations, dependent variable is of binary nature. It takes the value 1 in Eq (1) if a household is caloric poor and 0 otherwise and used the same cut point of 2350Kcal per adult equivalent/day across the years and region. In Eq (4), household takes the value 1 and identified as poor if a household expenditure per adult equivalent per month is less than the official poverty line (OPL) in the respective year and 0 otherwise for non-poor. To capture the effect of price, the main variable of interest, on the probability of a household being in poverty, we include the prices of 11 food groups by adding a set of variables in the analysis. The selection of rest of control variables is guided by the economic theory and by empirical perspective. The standard variables such as the household characteristics, educational level of household head and occupation type of household plays an important role in determining poverty and to get the estimates of poverty in both Pakistan and other countries as well [23]. To analyze the effect of occupation on the probability of a household being poor, we use dummy variable corresponding to two occupational groups such as employed in agricultural activity and non-agricultural activity as a base category. With respect to education of household head, we include dummy variables corresponding to not literate, primary level, higher secondary level and having education graduate and above. The base category in our analysis is that household head is not literate. We also used dummy variables for the other environmental variables such as water and sanitation, livestock, for holding of assets such as having agriculture land, non-agriculture land, residential property and commercial property. In addition to the above independent variables, we have incorporated number of background and demographic variables. Among demographic variables, we used age of the household head (in years), female headed household which usually considered being more likely in poverty, household size, female proportion in a household and composition of household that takes into account the proportion of male and female of different age brackets.

## Results and discussion

### Descriptive analysis

Table 1 is displaying the above mentioned FGT poverty index for the year 2005–06, 2007–08 and 2010–11 as well as for rural and urban areas for these survey years.

The official poverty line in case of income poverty was Rs. 1277.47, Rs. 1543.51 and Rs. 2333.35 per adult equivalent per month in 2005–06, 2007–08 and in 2010–11 respectively. However, the same calorie norms or a cutoff point that is 2350 kcal per adult equivalent per day was used for the three years to compute the poverty estimates. Table 1 shows a higher poverty incidence in 2005–06. A comparison between poverty estimates shows a declining trend in poverty incidence from 2005–06 to 2010–11 in case of expenditure poverty. This result is consistent with report of Pakistan Economic survey of year 2021[24]. While on the other side calorie poverty incidence is little bit increasing from 2005–06 to 2007–08 and then decreasing from 2007–08 to 2010–11 and more or less same from 2005–2010. In absolute terms, HCR of calorie poverty are far higher than income poverty. Result in Table 1 indicates that incidence of poverty based on income is higher in rural areas than their counterpart urban areas. In

**Table 1. Poverty estimates based on expenditure and calorie poverty line for the three years.**

| Years | Poverty Incidence (HCR) | | Poverty Depth (PG Index) | | Poverty Severity (FGT2 Index) | |
|---|---|---|---|---|---|---|
| | Income Based | Calorie Based | Income Based | Calorie Based | Income Based | Calorie Based |
| ALL | | | | | | |
| 2005–06 | 55.59 | 65.74 | 3.92 | 15.81 | 1.1 | 5.45 |
| 2007–08 | 44.54 | 66.9 | 2.74 | 15.84 | 0.7 | 5.28 |
| 2010–11 | 36.47 | 64.63 | 1.81 | 13.95 | 0.42 | 4.26 |
| URBAN | | | | | | |
| 2005–06 | 36.28 | 72.38 | 1.2 | 18.95 | 0.56 | 6.8 |
| 2007–08 | 32.88 | 72.11 | 1.34 | 18.39 | 0.32 | 6.5 |
| 2010–11 | 26.25 | 72.6 | 1.1 | 16.40 | 0.27 | 5.2 |
| RURAL | | | | | | |
| 2005–06 | 57.56 | 61.3 | 5.2 | 13.68 | 1.4 | 4.00 |
| 2007–08 | 49.58 | 63.38 | 3.7 | 14.12 | 0.96 | 4.5 |
| 2010–11 | 42.78 | 59.90 | 2.3 | 12.30 | 0.52 | 3.6 |

contrast; the incidence of calorie poverty is higher in urban areas than rural areas. Similar trends exist in the depth and severity of poverty measures for urban and rural areas across the years.

The calories $P(1)$ and $P(2)$ are higher in all three survey years than income $P(1)$ and $P(2)$. The lower values of $P(1)$ and $P(2)$ for income poverty shows that poor individuals are bunched around the poverty line while higher values for calorie poverty indicates the remoteness of poor individuals from the poverty line. In 2005–06 calorie poverty is nearly five times higher than those based on income. Despite an increase in poverty incidence based on calorie poverty its depth is declining in 2010. Poverty severity is decreasing with higher magnitude in income poverty than calorie poverty. Thus, in contrast to trends in income poverty, where direction of change appears largely invariant to the choice of FGT, trends in calorie poverty are sensitive to this choice. It is very hard to believe that poverty has decreasing trend in Pakistan over time. However, if we look at the calorie poverty than we come to know that a significant number of people in Pakistan are still holding the poverty status. Hence, we can conclude from the poverty measures that income poverty in Pakistan is devastatingly a rural phenomenon which is consistent with the study [25] in literature, while calorie poverty is crushingly urban phenomenon.

In order to further classify the population under different poverty bands, we estimate the poverty bands for both income and calorie poverty in Table 2. Poverty bands can be defined as "the fraction of population living at various distances from the poverty line and expressed in terms of rupees above or below the poverty line" [14]. The GOP (2008) used 6 poverty bands such as extremely poor; individuals consuming less than 50 percent of poverty line expenditures, decreased from 0.46 to 0.08 and 4.3 to 1.8 percent of population based on income and calorie poverty respectively. The second poverty band that is ultra-poor; comprising the population that are consuming less than 75 and more than 50 percent of poverty line indicates an improvement of 3.38 percentage points for income poverty and 1.4 percentage points for calorie poverty. At the other side of band, fifth poverty band "Quasi Non poor" indicates an increase from 34.17 percent to 40.88 percent in case of income poverty while more or less same for calorie poverty. The fourth poverty band named as vulnerable; comprising less than 125 and more than 100 percent of poverty line, has decreasing trend for income poverty while increasing trend for calorie poverty. It seems that any unusual or personal shock can easily

**Table 2. Percentage of population under poverty bands for all three years.**

| Poverty Bands | 2005–06 | | 2007–08 | | 2010–11 | |
|---|---|---|---|---|---|---|
| | Income Poverty | Calorie Poverty | Income Poverty | Calorie Poverty | Income Poverty | Calorie Poverty |
| ALL | | | | | | |
| Extremely Poor | 0.46 | 4.3 | 0.15 | 3.6 | 0.08 | 1.8 |
| Ultra Poor | 5.44 | 23.9 | 3.83 | 25.3 | 2.06 | 22.5 |
| Poor | 15.69 | 37.5 | 12.56 | 38.00 | 10.34 | 40.3 |
| Vulnerable | 18.1 | 21.3 | 17.83 | 20.7 | 16.5 | 22.8 |
| Quasi Nonpoor | 34.17 | 11.9 | 36.64 | 11.6 | 40.88 | 11.8 |
| Non-Poor | 26.13 | 1.00 | 28.99 | 0.8 | 30.14 | 0.7 |
| URBAN | | | | | | |
| Extremely Poor | 0.002 | 0.05 | 0.001 | 0.05 | 0.001 | 0.03 |
| Ultra Poor | 0.03 | 0.30 | 0.02 | 0.29 | 0.01 | 0.27 |
| Poor | 0.10 | 0.36 | 0.07 | 0.37 | 0.06 | 0.42 |
| Vulnerable | 0.13 | 0.18 | 0.13 | 0.18 | 0.12 | 0.19 |
| Quasi Nonpoor | 0.33 | 0.09 | 0.34 | 0.09 | 0.37 | 0.09 |
| Non-Poor | 0.41 | 0.01 | 0.44 | 0.09 | 0.43 | 0.004 |
| RURAL | | | | | | |
| Extremely Poor | 0.01 | 0.03 | 0.002 | 0.03 | 0.001 | 0.01 |
| Ultra Poor | 0.07 | 0.20 | 0.053 | 0.22 | 0.03 | 0.19 |
| Poor | 0.19 | 0.38 | 0.16 | 0.39 | 0.13 | 0.39 |
| Vulnerable | 0.21 | 0.24 | 0.21 | 0.23 | 0.19 | 0.25 |
| Quasi Nonpoor | 0.35 | 0.14 | 0.38 | 0.13 | 0.43 | 0.14 |
| Non-Poor | 0.16 | 0.01 | 0.19 | 0.01 | 0.21 | 0.01 |

shift the households into the category of poor in case of income poverty. By taking together, the category poor with vulnerable, the poverty status of population is prone to swing with food inflation and growth performance of agriculture sector in the country [24].

## Empirical analysis

**Calorie poverty model.** Complete results for logit estimation are given in Table 3 but we discuss only the price elasticities in detail. All the models fit the data well and the combined effect of regional, socioeconomic and environmental variables are significant at 5% and 10% level of significance, respectively. Results in Table 3 indicate that price of all food groups are significant in 2005–06 for calorie-based model except the prices of vegetables and spices that are insignificant. The estimated price responses of some important food groups such as milk, meat, fruit, spices, wheat, rice, pulses and other foods are significant and have expected signs. Wheat is the staple food in Pakistan and people usually get on average more than 50% of their calories from wheat [20,26]. An increase in the price of wheat increased the probability of a household being in calorie poverty by 77% in 2005–06, whereas effect price of wheat in 2007–08 is negative and significant and again in 2010–11 it has positive and significant effect on poverty. Wheat data has a great deal of spatial and temporal variation. When prices rise disproportionately among food groups, households substitute cheaper foods with more expensive foods to cope up with prices and keep the calorie level more or less constant. Although the substitution mitigates the negative effect of food prices for poor households than non-poor households but distributional consequences does not change. Still, poor households need to spend larger portion of their income on food. In calorie poverty model, the prices of milk, meat, sugar, wheat, rice, pulses and other foods are significant and show that with a 1% increase in prices of

**Table 3. Calorie poverty model.**

| Variables | 2005–06 | | | 2007–08 | | | 2010–11 | | |
|---|---|---|---|---|---|---|---|---|---|
| | Parameter | T ratio | Elasticity | Parameter | T ratio | Elasticity | Parameter | T ratio | Elasticity |
| Log_MILKPrice | 0.475* | 2.59 | 0.097 | 0.703* | 4.65 | 0.139 | 0.328 | 1.62 | 0.066 |
| Log_MEATPrice | 0.106 | 1.04 | 0.022 | -1.688* | -12.88 | -0.333 | -0.980* | -8.73 | -0.199 |
| Log_FRUITPrice | 0.267* | 2.6 | 0.055 | -0.156 | -1.43 | -0.031 | 0.002 | 0.03 | 0.001 |
| Log_VEGPrice | -0.318 | -0.83 | -0.065 | 3.102* | 11.19 | 0.612 | 2.393* | 7.6 | 0.485 |
| Log_SPICESPrice | -0.153 | -2.47 | -0.031 | -0.208* | -2.84 | -0.041 | -0.032 | -0.45 | -0.007 |
| Log_SUGARPrice | -0.788* | -4.92 | -0.161 | 0.176 | 1.25 | 0.035 | -0.329* | -1.99 | -0.066 |
| Log_WHEATPrice | 3.772* | 9.15 | 0.771 | -1.231* | -4.29 | -0.243 | 0.824 | 1.37 | 0.167 |
| Log_RICEPrice | 0.935* | 5.83 | 0.191 | 0.569* | 2.63 | 0.113 | 1.978* | 9.2 | 0.401 |
| Log_PULSESPrice | 1.018* | 2.77 | 0.208 | -0.718 | -1.58 | -0.142 | 0.896* | 2.3 | 0.182 |
| Log_OILPrice | -0.240 | -0.71 | -0.049 | -2.169* | -5.3 | -0.428 | -4.490* | -8.18 | -0.911 |
| Log_OTHERFOODSPrice | 0.128* | 3.06 | 0.026 | 0.166* | 3.21 | 0.033 | 0.322* | 8.38 | 0.065 |
| Log_PCME | -2.669* | -42.89 | -0.545 | -3.120* | -43.67 | -0.616 | -3.724* | -49.06 | -0.755 |
| HHsize | 0.099* | 11.62 | 0.020 | 0.152* | 12.92 | 0.029 | 0.082* | 6.28 | 0.017 |
| F_HHH | -0.068 | -0.78 | -0.014 | -0.011 | -0.13 | -0.002 | -0.094 | -1.09 | -0.019 |
| Age_HHH | -0.0001 | -0.09 | -0.0003 | -0.002 | -0.98 | -0.0004 | 0.001 | 0.66 | 0.0003 |
| Head_Primary | -0.090* | -1.99 | -0.018 | -0.014 | -0.3 | -0.003 | 0.011 | 0.22 | 0.002 |
| Head_HigherSecondary | 0.098 | 0.87 | 0.020 | 0.103 | 0.92 | 0.020 | 0.264* | 3.33 | 0.054 |
| Head_Graduate | 0.708* | 8.42 | 0.128 | 0.490 | 5.81 | 0.088 | 0.388* | 5.82 | 0.074 |
| E_Status | -0.714* | -13.79 | -0.155 | -0.817* | -14.47 | -0.175 | -0.783* | -13.42 | -0.172 |
| FemaleProportion | 0.049** | 2.05 | 0.010 | 0.025 | 0.6 | 0.005 | 0.063* | 5.2 | 0.013 |
| WomenEducation | -0.011 | -0.09 | -0.002 | 0.107 | 0.89 | 0.021 | -0.027 | -1.18 | -0.005 |
| rm0_4 | 0.039 | 1.64 | 0.008 | 0.014 | 0.34 | 0.003 | 0.049* | 4.13 | 0.010 |
| rm5_9 | 0.061* | 2.52 | 0.013 | 0.035 | 0.85 | 0.007 | 0.071* | 5.92 | 0.014 |
| rm10_14 | 0.069* | 2.86 | 0.014 | 0.043 | 1.03 | 0.008 | 0.076* | 6.35 | 0.015 |
| rm15_55 | 0.080* | 3.3 | 0.016 | 0.058 | 1.4 | 0.011 | 0.089* | 7.59 | 0.018 |
| rm56P | 0.069* | 2.83 | 0.014 | 0.046 | 1.11 | 0.009 | 0.079* | 6.57 | 0.016 |
| rf0_4 | -0.009* | -3.04 | -0.002 | -0.008* | -2.32 | -0.001 | -0.013* | -4 | -0.003 |
| rf5_9 | 0.002 | 0.7 | 0.0004 | 0.0003 | 0.08 | 0.000 | 0.004 | 1.21 | 0.001 |
| rf10_14 | 0.002 | 0.8 | 0.0005 | 0.004 | 1.29 | 0.001 | 0.001 | 0.28 | 0.002 |
| rf15_55 | -0.004 | -1.78 | -0.0009 | -0.003 | -1.25 | -0.001 | -0.000 | -0.09 | -0.0004 |
| Agri_LandOwner | -0.159* | -2.26 | -0.033 | -0.118 | -1.59 | -0.024 | -0.165* | -2.07 | -0.034 |
| NonAgri_LandOwner | -0.296* | -2.68 | -0.064 | -0.050 | -0.36 | -0.009 | -0.032 | -0.22 | -0.007 |
| Residential_Property | -0.038 | -0.56 | -0.008 | -0.096 | -1.33 | -0.019 | 0.137* | 2.29 | 0.028 |
| Commercial_Property | 0.011 | 0.11 | 0.002 | -0.072 | -0.57 | -0.014 | -0.267* | -2.29 | -0.057 |
| Livestock | -0.214* | -2.9 | -0.045 | -0.333* | -3.83 | -0.069 | -0.056 | -0.62 | -0.011 |
| Access_Water | -0.055 | -0.8 | -0.011 | -0.089 | -1.32 | -0.017 | 0.127* | 1.94 | 0.026 |
| No_Toilet | -0.250* | -4.41 | -0.052 | -0.396* | -6.64 | -0.082 | -0.198* | -2.97 | -0.041 |
| Urban | 0.709* | 10.9 | 0.140 | 0.682* | 9.45 | 0.129 | 0.611* | 9.32 | 0.121 |
| Sindh | 0.184 | 1.76 | 0.037 | 1.354* | 10.91 | 0.223 | 0.253* | 2.12 | 0.049 |
| KPK | -0.431* | -3.58 | -0.093 | -0.412* | -4.36 | -0.086 | -0.851* | -6.21 | -0.189 |
| Baluchistan | -0.438* | -3.7 | -0.095 | -0.069 | -0.51 | -0.014 | -0.712* | -4.14 | -0.158 |
| Constant | -0.489 | -0.15 | | 29.774 | 6.11 | | 26.461 | 6.89 | |
| No. of observations | 15412 | | | 14983 | | | 15616 | | |
| Log-likelihood ratio | -7151.2258 | | | -6570.0304 | | | -6643.2001 | | |

All elasticities are evaluated at sample means. Parameter estimates with (*) signs are significant at 5% level of significance.

these food groups increase the probability of a household in poverty from 2.6% to 77% respectively. Over time, increase in the prices of different food groups affect the households differently primarily because of their relative importance in the household budget.

The poor households spend more of their food budget on wheat, spices, vegetables, oil and sugar and spend proportionally less on milk, meat and fruit. Milk is the imperative food group in our diet. Increase in the price of milk will push the household into calorie poverty in all three years with probability of 6% to 14% across survey period. It can be viewed as that household are cutting back their calories from milk and spending that amount on the substitution of milk. Price of meat has significant and positive effect on poverty status in 2005–06 while significant and negative effect in 2007–2010. The price of vegetables has significant and negative effect in 2005–06 while it has positive association with poverty status across 2007–2010. The estimated price elasticity is positive and significant for rice and pulses in calorie-based poverty. However, the probability of shifting a household from non-poor to poor falls with oil price, implying that oil is the complementary item to prepare a meal. Household can switch from better quality to lower quality of a food group but cannot give up the use of some essential commodities. Thus, the effect of rising food prices, especially the price of milk, meat, fruit, wheat, rice, pulses appear to distinguish the status of a household.

### Urban and rural differences (Calorie poverty model)

The impact of the price increases on poverty varies among countries and within country. Results in Table A.1 and A.2 in S1 File indicates that price of meat, wheat, pulses and other food groups have significant effect in shifting the household in poverty state in both urban and rural areas in 2005–06. The estimated price elasticity of wheat increases the probability of a household being in poverty from 46% to 115% from 2005–2010. It is insignificant for rural population from 2007–2010. Similarly, the increase in the price of rice hit the urban households from 0.8% to 60% across the survey years. For rural households it is insignificant in the first two survey rounds while have significant effect in 2010–11. These price elasticities (2005–2010) show that urban households are hit harder than rural households in calorie poverty model. Although many households in rural areas are also net consumers of food, and therefore, adversely affected by price increase [11].

### Income poverty model

Results in Table 4 indicate that price of all food groups are significant in 2005–06 in income-based model. An increase in the price of wheat increased the probability of a household being in poverty by 17% in 2005–06, while poverty is not significantly influenced by the price of wheat in 2007–08 and increasing the probability of household moving out of poverty by 18%. Expenditure poverty model in 2005–06 indicates that probability of a household being in poverty is almost same for sugar (17%) and wheat (17%). Price of rice is significant and negative in all survey years. It can be viewed as that probability of a household falling into poverty is decreasing with a 1% increase in price of rice. Meat price is increasing the probability of household moving into poverty by 2% to 5% from 2005/06 to 2007/08, whereas insignificant in 2010/11.

Price of pulses is contributing to increase the probability of a household falling into poverty across years from 25%, 21% and 9%, respectively. Impact of price of fruit and vegetables are negative and significant. Fruit prices effect is somewhat puzzling as it is common that poor household do not consume fruit as much as rich household, while poor household consume more vegetables in their diet. A negative price elasticity of vegetable in 2005/06 and 2007/08 suggest that it is not increasing the probability of a household being in poverty, people

**Table 4. Income poverty model.**

| Variables | 2005–06 | | | 2007–08 | | | 2010–11 | | |
|---|---|---|---|---|---|---|---|---|---|
| | Parameter | T ratio | Elasticity | Parameter | T ratio | Elasticity | Parameter | T ratio | Elasticity |
| Log_MILKPrice | -0.944* | -4.67 | -0.114 | -0.081 | -0.44 | -0.006 | -1.651* | -6.58 | -0.104 |
| Log_MEATPrice | 0.489* | 4.25 | 0.058 | 0.310* | 1.92 | 0.022 | -0.042 | -0.3 | -0.003 |
| Log_FRUITPrice | -0.686* | -6.69 | -0.083 | -0.760* | -6.2 | -0.054 | -0.122 | -1.27 | -0.008 |
| Log_VEGPrice | -1.984* | -4.63 | -0.239 | -0.956* | -2.75 | -0.068 | -1.219* | -3.28 | -0.077 |
| Log_SPICESPrice | 0.406* | 6.1 | 0.049 | -0.032 | -0.36 | -0.002 | 0.236* | 2.7 | 0.015 |
| Log_SUGARPrice | 1.436* | 7.71 | 0.173 | 2.169* | 10.61 | 0.154 | -0.086 | -0.4 | -0.005 |
| Log_WHEATPrice | 1.480* | 3.27 | 0.178 | -0.176 | -0.45 | -0.013 | -2.778* | -4 | -0.176 |
| Log_RICEPrice | -0.905* | -5.29 | -0.109 | -0.898* | -3.22 | -0.064 | -1.239* | -4.74 | -0.078 |
| Log_PULSESPrice | 2.087* | 5.13 | 0.252 | 3.030* | 5.61 | 0.215 | 1.533* | 3.09 | 0.097 |
| Log_OILPrice | -1.227* | -3.19 | -0.148 | -0.101 | -0.21 | -0.007 | -6.753* | -8.78 | -0.427 |
| Log_OTHERFOODSPrice | 0.241* | 5.29 | 0.029 | -0.092 | -1.5 | -0.007 | -0.021 | -0.46 | -0.001 |
| HHsize | 0.209* | 26.72 | 0.025 | 0.392* | 31.69 | 0.028 | 0.274* | 20.19 | 0.017 |
| F_HHH | -0.608* | -5.03 | -0.061 | -0.136 | -1.1 | -0.010 | -0.544* | -4.22 | -0.028 |
| Age_HHH | 0.007* | 3.06 | 0.001 | 0.004 | 1.36 | 0.0002 | -0.003 | -1.05 | -0.0001 |
| Head_Primary | -0.014 | -0.29 | -0.002 | -0.083 | -1.5 | -0.006 | -0.163* | -2.41 | -0.010 |
| Head_HigherSecondary | -0.219 | -1.08 | -0.026 | -0.071 | -0.27 | -0.005 | -0.552* | -5.04 | -0.038 |
| Head_Graduate | -1.536* | -11.13 | -0.126 | -1.735 | -9.54 | -0.078 | -1.469* | -11.4 | -0.065 |
| E_Status | -0.647* | -11.68 | -0.070 | -0.653 | -10.1 | -0.041 | -1.071* | -13.76 | -0.053 |
| FemaleProportion | 0.035 | 1.43 | 0.004 | 0.134* | 3.5 | 0.009 | 0.037* | 3.98 | 0.002 |
| WomenEducation | -0.527* | -2.56 | -0.065 | -0.746 | -2.82 | -0.057 | -0.043 | -1.65 | -0.003 |
| rm0_4 | 0.032 | 1.31 | 0.004 | 0.130* | 3.39 | 0.009 | 0.028* | 3.26 | 0.002 |
| rm5_9 | 0.049* | 2.02 | 0.006 | 0.149* | 3.88 | 0.011 | 0.055* | 6.38 | 0.003 |
| rm10_14 | 0.056* | 2.29 | 0.007 | 0.147* | 3.81 | 0.010 | 0.059* | 6.8 | 0.004 |
| rm15_55 | 0.042 | 1.7 | 0.005 | 0.144* | 3.74 | 0.010 | 0.038* | 4.62 | 0.002 |
| rm56P | 0.031 | 1.26 | 0.004 | 0.138 | 3.57 | 0.009 | 0.037* | 3.9 | 0.002 |
| rf0_4 | -0.002 | -0.42 | -0.000 | -0.002 | -0.48 | -0.0002 | -0.015* | -2.92 | -0.010 |
| rf5_9 | 0.010* | 2.55 | 0.001 | 0.004 | 1.04 | 0.0004 | 0.014* | 2.98 | 0.001 |
| rf10_14 | 0.018* | 4.62 | 0.002 | 0.007 | 1.62 | 0.0006 | 0.019* | 4.03 | 0.001 |
| rf15_55 | -0.004 | -1.27 | -0.001 | -0.008 | -1.8 | -0.0006 | -0.005 | -1.09 | -0.0003 |
| Agri_LandOwner | -0.667* | -7.1 | -0.066 | -0.668 | -6 | -0.038 | -0.797* | -5.98 | -0.038 |
| NonAgri_LandOwner | -0.122 | -0.99 | -0.014 | 0.259 | 1.68 | 0.020 | -1.057* | -4.11 | -0.044 |
| Residential_Property | -0.329* | -4.66 | -0.044 | -0.402* | -4.66 | -0.032 | -0.426* | -5.9 | -0.030 |
| Commercial_Property | -0.554* | -3.77 | -0.055 | -1.077* | -4.3 | -0.051 | -0.829* | -3.88 | -0.038 |
| Livestock | 0.234* | 3.1 | 0.030 | 0.244* | 2.63 | 0.019 | -0.436* | -3.65 | -0.023 |
| Access_Water | -0.112 | -1.66 | -0.014 | 0.195* | 2.47 | 0.013 | -0.194* | -2.43 | -0.013 |
| No_Toilet | 0.837* | 14.61 | 0.117 | 0.916* | 14.32 | 0.082 | 0.599* | 8.18 | 0.046 |
| Urban | -0.276* | -3.95 | -0.033 | -0.343* | -3.9 | -0.024 | -0.421* | -5.25 | -0.026 |
| Sindh | -0.358* | -2.88 | -0.041 | -0.425* | -2.68 | -0.028 | -0.511* | -3.47 | -0.029 |
| KPK | 0.194 | 1.37 | 0.025 | -0.033 | -0.26 | -0.002 | 0.122 | 0.68 | 0.008 |
| Baluchistan | 0.376* | 2.78 | 0.050 | 0.573* | 3.52 | 0.048 | -1.132* | -5.37 | -0.052 |
| Constant | -5.969 | -1.79 | | -29.116* | -5.89 | | 46.580* | 9.86 | |
| No. of observations | 15453 | | | 14983 | | | 15632 | | |
| Log-likelihood ratio | -6158.6691 | | | -4820.8683 | | | -4579.0064 | | |

All elasticities are evaluated at sample means. Parameter estimates with (*) signs are significant at 5% level of significance.

consume their self-produced vegetables and also sell the surplus as a consequence of increased in prices. It can be viewed as these households are consuming less food than their production to get benefit for the higher price in market.

## Urban and rural differences (Income poverty model)

Rising food prices aggravate inequality between poor and rich households. Moreover, households that currently live just above the poverty line may fall into poverty as a result of food price increases. The meat price is increasing the chance of urban households being in poverty in all survey years. On the other hand, price of milk has significant and but negative impact and seems to suggest that estimated elasticity of milk price is not increasing the probability of a household falls in poverty both in urban and rural areas. This result is consistent with the study of Shabnam et al. [7].

Price of wheat is not significant in 2005/06 in urban areas while increasing the incidence of poverty in rural areas by 26%, 24% in 2007/08 and then decreasing the probability being in poverty by 18% in 2010/11. The estimated price elasticity for another staple food-rice- is also negative in both urban and rural areas. Pakistan is a net exporter of rice and rice forms a small component, about 4% to 5% of food expenditures, of domestic consumption [6]. With the increase in price of sugar both urban and rural households have greater chance to shift in poverty state in 2005/06 and 2007/08, because they will spend more budget to buy the specific amount while it is insignificant in 2010/11. The estimated price elasticities of essential food groups are greater than 1, which suggest that either the more and closer substitute are available so that people can easily switch from one food group to other or we can say that they have stronger substitution effect. Similarly, for most food groups, the longer a price change holds, the higher the elasticity is, as more and more consumers start inclination to search for substitutes. It appears that being poor is not a random occurrence.

There is distinct demographic, social and economic factors that can force a household into a state of poverty. The appropriate assessment of price effect has been somewhat shaky. Also, there are many channels which may influence the price effect [27]. These insights can potentially be very useful in the design of a poverty alleviation program [28]. Despite an apparent divergence between calorie-based poverty and income-based poverty trends, income continues to be a powerful determinant of calorie intake [29,30]. Household income plays an important role in moving out from poverty state [26]. With the increase in income, household will be able to get better quality food and consume more calories. The effect of household income is found to be negative and significant from 2005–2010 as well as in urban and rural areas. As consistent with earlier studies, our analysis show that larger household size is increasing the probability of a household moving into poverty in both poverty models as well as with respect to rural urban areas [31]. Gender of the head is insignificant in calorie-based model, whereas have significant and negative effect on income-based poverty. The study] 31] also reported that the 67% of households headed by female did not experience the poverty and are 48% are managing much better than male-headed households. So, we cannot compel our decision to the point that female headed households should be worse off than male headed households. Age of the household head is negative and significant in rural areas in 2005–2007 in calorie model; suggest that initially the household head empowers the household through enhancing the economic activity to prevent the household moving into poverty state.

Household head education levels have somewhat puzzling effect. The education of a household head has a significant and negative association with poverty in income poverty model, both in rural and urban areas, suggesting that households headed by literate persons are less likely to be poor than headed by illiterate persons. On the other hand, education of head is not

supporting the notion that it plays a significant role in the taking the decision regarding calorie consumption. The coefficient is positive at higher level of education in calorie poverty model. It might be the case that educated household head spend more on non-food items than food items. It is not an easy to put in plain words this trend since education is considered an imperative factor in helping the households moving out of poverty.

Employment status has negative and significant effect in almost all models. In terms of poverty, it suggests that households engaged in agricultural activities are poor than non-agricultural households. An educated woman can mitigate the chances of a household towards poverty by contributing through earning as well as play an important role in determining food consumption, and thus the calorie consumption per capita, of household members. To examine the impact of this factor on calorie and income poverty we include the average years of schooling from adult women in the family. Result shows that woman education decreased the probability of a household falls into poverty in urban and rural areas. Thus, an educated female household mitigate the effect of poverty. Relatively few of the household composition effect are significant in urban and rural households. In income-based poverty model, household level assets such as ownership of agriculture land, non-agriculture land, residential property and commercial property have a significant (negative) association with poverty status in rural urban disaggregation. In calorie poverty model, holding agriculture property has significant and negative association whereas relatively few observations are significant for other assets. In 2010/11, ownership of livestock in rural area has significant (negative) association. On the other side, in calorie poverty model, livestock has negative and significant association with the poverty status of a household.

The environmental hygiene variables such as access to improved drinking water and toilet facility in a household generally have negative and significant association with poverty status. In these both poverty models, households having access to improve drinking water is decreasing the probability of a household move towards poverty. Not having the toilet facility in a household increased the probability of household being in poverty.

Our result of bivariate analysis is consistent with the results of poverty measure that income-based poverty is rural phenomenon while calorie poverty is related to urban. To capture the location effect on poverty, a dummy variable for province is introduced. Punjab is the base category. Income poverty is decreasing in Sindh than Punjab while calorie poverty is decreasing in KPK across survey years. The analysis underscores the vulnerable situation in Baluchistan.

## Conclusion

These findings potentially point to some important implications. Most importantly, they suggest that international food price movements might be important drivers of national and global poverty trends as well as rural-urban inequality within countries. In this paper we have analyzed the impact of food price crisis on calorie and income poverty in Pakistan, as well as explored the poverty correlates in the context of rural and urban areas for the period of 2005–2010. Increase in food prices hurts the poor a lot as more than 50% are net buyers and belongs to rural areas. It is necessary to look at the consumption pattern of household to examine the impact of price increase. Rising global and domestic price affected Pakistani households throughout its increase.

The study used three survey rounds, 2005–06, 2007–08 and 2010–11. These rounds have also been used as cross-sectional datasets to examine trends in poverty. We have estimated poverty using the official poverty line across the years. The poverty measures show a fluctuation for calorie poverty, increase in 2007–08 and then decrease in 2010–11, whereas there is a

declining trend in income poverty over survey years [24]. Analysis revealed that income poverty is clearly a rural phenomenon, around 16% rural households are poor while calorie poverty is urban phenomenon with 73% poor households residing in urban areas. Urban households consumed fewer calories on average than rural households. This decline is considered to be direct result of rising food prices, due to fewer market purchases of various categories of food. The estimates show that 2.9 million people are unable to meet the 50 percent of expenditure of poverty line in 2005–06. This number has decreased from 2005 to 2010 in terms of income poverty and calorie poverty, although the decreasing rate is higher for income poverty than calorie poverty. Significance differences exist in the incidence of poverty across the country. It can be concluded that, if prices continue rising, they could become vital lifelines for millions of vulnerable families in case of calorie poverty. The findings from empirical analysis shows that prices of essential food groups have positive association with income and calorie poverty. Agriculture is like a backbone of Pakistan's economy and takes on around 47 percent of population in this sector. The population of country is growing rapidly and being a net importer of food, especially wheat, poverty alleviation is big challenge for policy makers. Analysis shows that, an increase in price of wheat increased the probability of household to move in calorie poverty in 2005–06 and 2010–11. On the other side, increase in wheat prices is not increasing the poverty in rural areas in income poverty model. However, a small proportion of households are net producer of wheat and larger proportion is net consumer in a country. Previous studies show that increase in wheat prices has worsen effect on urban households than rural [7,20]. It might be agriculture households sought to capitalize on the relatively higher prices for agriculture goods, although more work is required before this conclusion can be drawn with confidence [32,33]. Price of another staple food such as rice has positive and significant association with calorie poverty as well as negative and significant association with income poverty over urban and rural areas respectively. Pakistan is a net exporter of rice and rice forms a small component, about 4% to 5% of food expenditures, of domestic consumption and considered to be cash crop in Pakistan. It seems as rice price is not increasing the probability of a household's being in poverty. The empirical finding indicates that there are some other demographic and economic factors that affect the status of a household. The poor have usually larger families, lower education level, having no access to land, limited access to supplementary income sources. These all-demographic burden increases the probability of a household falling into poverty. Economic factors such as income, possession of physical assets (agriculture land, commercial, residential) improves households wellbeing in rural and urban households.

## Policy implication and recommendations

Food price crisis in Pakistan broadened the gap between the poor and non-poor and exacerbated inequalities across region. In the short run, it is imperative to ensure food availability to vulnerable households to help them escape from calorie deprivation. The government could raise the poverty cut-off score or complement Benazir Income Support Program's (BISP) current targeting scheme with some more targeting strategies. For example, the government could identify households at different poverty thresholds that lack access to land. Aside from cash transfer programs such as BISP, the government could consider supply interventions that make food more accessible, particularly for households without access to agricultural land. For instance, the government maintains networks of utility stores to sell discounted food but places no limitations on who can purchase that food. The government may want to consider targeting eligibility to purchase discounted food to poorer segments of the population, identified on the basis of the poverty scorecard census. In the longer run, high and sustainable economic growth rate will serve the purpose of making people food secure and to escape poverty. The

unrelenting protection of consumers by maintaining local food prices, particularly wheat prices, below international prices deter increases in domestic supply. However, safety nets will be necessary for the most vulnerable households as they won't be able to support their livelihoods if global food prices continue to rise and the government decides to raise domestic prices to international rates. Moreover, there is need to broaden the scope of income or food support program for the benefit of the poor.

## Supporting information

**S1 File.**
(ZIP)

## Author Contributions

**Conceptualization:** Nadia Shabnam, Salma Riaz.

**Formal analysis:** Nadia Shabnam, Neelam Aurangzeb, Salma Riaz.

**Investigation:** Nadia Shabnam.

**Methodology:** Nadia Shabnam, Neelam Aurangzeb, Salma Riaz.

**Software:** Nadia Shabnam, Neelam Aurangzeb, Salma Riaz.

**Supervision:** Nadia Shabnam.

**Validation:** Nadia Shabnam.

**Visualization:** Nadia Shabnam.

**Writing – original draft:** Nadia Shabnam, Neelam Aurangzeb.

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
