## [Decision Letter · Decision Letter 0]

9 Mar 2023

PONE-D-23-03624RISING FOOD PRICES AND POVERTY IN PAKISTANPLOS ONE

Dear Dr. Shabnam,

Thank you for submitting your manuscript to PLOS ONE. After careful consideration, we feel that it has merit but does not fully meet PLOS ONE’s publication criteria as it currently stands. Therefore, we invite you to submit a revised version of the manuscript that addresses the points raised during the review process.

We look forward to receiving your revised manuscript.

Kind regards,

Faisal Abbas, PhD

Academic Editor

PLOS ONE

Journal Requirements:

3. Please ensure that you include a title page within your main document. You should list all authors and all affiliations as per our author instructions and clearly indicate the corresponding author.

4. We note you have included a table to which you do not refer in the text of your manuscript. Please ensure that you refer to Table 1 in your text; if accepted, production will need this reference to link the reader to the Table.

**Additional Editor Comments:**

Minor Revision.

Reviewers' comments:

Reviewer's Responses to Questions

**Comments to the Author**

1. Is the manuscript technically sound, and do the data support the conclusions?

Reviewer #1: Yes

Reviewer #2: Yes

2. Has the statistical analysis been performed appropriately and rigorously? 

Reviewer #1: Yes

Reviewer #2: Yes

3. Have the authors made all data underlying the findings in their manuscript fully available?

Reviewer #1: Yes

Reviewer #2: Yes

4. Is the manuscript presented in an intelligible fashion and written in standard English?

Reviewer #1: Yes

Reviewer #2: Yes

5. Review Comments to the Author

Reviewer #1: Rising food prices and their impacts on poverty is a very pertinent issue for Pakistan these days. The paper attempts to explore this relationship in an articulated way. However, I have few minor suggestions.

1. The abstract of the paper needs to be improved.

2. Descriptive analysis should be explained in the results section rather than Methodology section.

3. In Tables 3 & 4, the variables need to be renamed for better understanding of the table.

Reviewer #2: Revise the abstract and make it more crisp and interesting.

A stronger motivation and economic intuition for the topic should be provided in introduction section. Scarcity does not mean necessity. You need to prove your paper is valuable enough and distinguish from others.

Please compare and contrast the relevant studies from existing literature in the results and discussion section.

Based upon your empirical results, propose concrete policy recommendations and provide future directions for researchers.

6. PLOS authors have the option to publish the peer review history of their article (what does this mean?). If published, this will include your full peer review and any attached files.

Reviewer #1: No

Reviewer #2: No

---

## [Author Response · Author response to Decision Letter 0]

15 Jun 2023

Journal Requirements:

Response: Thank you dear editor for pointing out these flaws in the formatting of manuscript. We have done all the style requirements in the revised manuscript.

2) In your Data Availability statement, you have not specified where the minimal data set underlying the results described in your manuscript can be found. PLOS defines a study's minimal data set as the underlying data used to reach the conclusions drawn in the manuscript and any additional data required to replicate the reported study findings in their entirety. All PLOS journals require that the minimal data set be made fully available. 

Response: We have provided the data in supporting file so that researcher can replicate the results. 

Please ensure that you include a title page within your main document. You should list all authors and all affiliations as per our author instructions and clearly indicate the corresponding author.

Response: Yes, we have included the title page with all the necessary requirements.

3) We note you have included a table to which you do not refer in the text of your manuscript. Please ensure that you refer to Table 1 in your text; if accepted, production will need this reference to link the reader to the Table.

Response: We have provided the reference of Table 1 in the body of the text.

Response: We have complete the reference list in the revised manuscript.

Reviewer # 1 (Review of the manuscript PONE-D-23-03624)

Rising food prices and their impacts on poverty is a very pertinent issue for Pakistan these days. The paper attempts to explore this relationship in an articulated way. However, I have few minor suggestions.

1) The abstract of the paper needs to be improved.

 Response: We are thankful to the reviewer for useful comments and highlighting this aspect. We have improved the abstract of the paper to our best.

2) Descriptive analysis should be explained in the results section rather than Methodology section.

Response: As per reviewer comment, we explained the descriptive analysis in the result section. Please see the line 198 to onward.

3) In Tables 3 & 4, the variables need to be renamed for better understanding of the table.

Response: As per reviewer comment, we have renamed the variables.

Reviewer # 2 (Review of the manuscript PONE-D-23-03624)

1) Revise the abstract and make it more crisp and interesting.

 RESPONSE: As per reviewer’s suggestion, we have revised the abstract.

2) A stronger motivation and economic intuition for the topic should be provided in introduction section. Scarcity does not mean necessity. You need to prove your paper is valuable enough and distinguish from others.

RESPONSE: We are really thankful to the reviewer for the valuable comment. We have incorporated the suggestions in introduction section. 

3) Please compare and contrast the relevant studies from existing literature in the results and discussion section.

 RESPONSE: As per reviewer comment, we have included the relevant studies for comparison in the discussion section.

4) Based upon your empirical results, propose concrete policy recommendations and provide future directions for researchers.

 RESPONSE: We have provided appropriate policy recommendations and future direction to the researchers. Please see the line 443 and section Policy implications and recommendation

---

## [Decision Letter · Decision Letter 1]

12 Sep 2023

RISING FOOD PRICES AND POVERTY IN PAKISTAN

PONE-D-23-03624R1

Dear Dr. Shabnam,

We’re pleased to inform you that your manuscript has been judged scientifically suitable for publication and will be formally accepted for publication once it meets all outstanding technical requirements.

Kind regards,

Gideon Kruseman, Ph.D.

Academic Editor

PLOS ONE

Additional Editor Comments (optional):

Reviewers' comments:

Reviewer's Responses to Questions

**Comments to the Author**

1. If the authors have adequately addressed your comments raised in a previous round of review and you feel that this manuscript is now acceptable for publication, you may indicate that here to bypass the “Comments to the Author” section, enter your conflict of interest statement in the “Confidential to Editor” section, and submit your "Accept" recommendation.

Reviewer #1: All comments have been addressed

Reviewer #2: (No Response)

2. Is the manuscript technically sound, and do the data support the conclusions?

Reviewer #1: Yes

Reviewer #2: (No Response)

3. Has the statistical analysis been performed appropriately and rigorously? 

Reviewer #1: Yes

Reviewer #2: (No Response)

4. Have the authors made all data underlying the findings in their manuscript fully available?

Reviewer #1: No

Reviewer #2: (No Response)

5. Is the manuscript presented in an intelligible fashion and written in standard English?

Reviewer #1: No

Reviewer #2: (No Response)

6. Review Comments to the Author

Reviewer #1: All comments have been addressed. Abstract has been revised, relevant literature is added. Policy recommendations are added

Reviewer #2: (No Response)

7. PLOS authors have the option to publish the peer review history of their article (what does this mean?). If published, this will include your full peer review and any attached files.

Reviewer #1: No

Reviewer #2: No

---

## [Editor Report · Acceptance letter]

18 Sep 2023

PONE-D-23-03624R1 

RISING FOOD PRICES AND POVERTY IN PAKISTAN 

Dear Dr. Shabnam:

I'm pleased to inform you that your manuscript has been deemed suitable for publication in PLOS ONE. Congratulations! Your manuscript is now with our production department. 

Kind regards, 

on behalf of

Dr. Gideon Kruseman 

Academic Editor

PLOS ONE